# Roman Urdu Hate Speech Detection Using Transformer-Based Model for Cyber Security Applications

**DOI:** 10.3390/s23083909

**Published:** 2023-04-12

**Authors:** Muhammad Bilal, Atif Khan, Salman Jan, Shahrulniza Musa, Shaukat Ali

**Affiliations:** 1Department of Computer Science, Islamia College Peshawar, Peshawar 25130, Pakistan; 2Malaysian Institute of Information Technology, Universiti Kuala Lumpur, Kuala Lumpur 50250, Malaysia; 3Department of Computer Science, Bacha Khan University Charsadda, Charsadda 24420, Pakistan

**Keywords:** BERT, BiLSTM, CNN, cyber security, deep learning, hate speech, LSTM, natural language processing (NLP), Roman Urdu, social media, transformer models

## Abstract

Social media applications, such as Twitter and Facebook, allow users to communicate and share their thoughts, status updates, opinions, photographs, and videos around the globe. Unfortunately, some people utilize these platforms to disseminate hate speech and abusive language. The growth of hate speech may result in hate crimes, cyber violence, and substantial harm to cyberspace, physical security, and social safety. As a result, hate speech detection is a critical issue for both cyberspace and physical society, necessitating the development of a robust application capable of detecting and combating it in real-time. Hate speech detection is a context-dependent problem that requires context-aware mechanisms for resolution. In this study, we employed a transformer-based model for Roman Urdu hate speech classification due to its ability to capture the text context. In addition, we developed the first Roman Urdu pre-trained BERT model, which we named BERT-RU. For this purpose, we exploited the capabilities of BERT by training it from scratch on the largest Roman Urdu dataset consisting of 173,714 text messages. Traditional and deep learning models were used as baseline models, including LSTM, BiLSTM, BiLSTM + Attention Layer, and CNN. We also investigated the concept of transfer learning by using pre-trained BERT embeddings in conjunction with deep learning models. The performance of each model was evaluated in terms of accuracy, precision, recall, and F-measure. The generalization of each model was evaluated on a cross-domain dataset. The experimental results revealed that the transformer-based model, when directly applied to the classification task of the Roman Urdu hate speech, outperformed traditional machine learning, deep learning models, and pre-trained transformer-based models in terms of accuracy, precision, recall, and F-measure, with scores of 96.70%, 97.25%, 96.74%, and 97.89%, respectively. In addition, the transformer-based model exhibited superior generalization on a cross-domain dataset.

## 1. Introduction

Social media applications, such as Twitter and Facebook, are the means through which individuals from all over the world communicate and share their thoughts, status updates, views, photos, and videos. Some individuals use it to promote abusive language and hate speech. Such hate speech can lead to severe repercussions, including acts of violence and assault.

Numerous individuals, especially women, are often subjected to online harassment. Some Internet users can become furious and feel endangered. Due to the toxic climate generated by racism, sectarian differences, intolerance, and hostility based on political and religious affiliations, many social media users have even abandoned social media platforms. Numerous governments across the globe have implemented legislation to prevent cybercrimes and the spread of hate speech in cyberspace, especially on social media networks. On the other side, social media companies are under immense pressure from the government and the public to combat hate speech and cyberbullying on their platforms.

Hate speech detection is an important issue that demands a powerful application to detect and combat it in real-time. Detecting hate speech is a context-dependent problem that requires context-aware models. Traditional models based on feature engineering could not efficiently recognize the context of a given text message. On the other hand, deep learning models, particularly RNN models with Word2Vec embeddings, performed well. However, there are certain limitations. RNN models work sequentially and suffer from long-term dependence. The Word2Vec model [1] assigns numerical values to each word in the corpus based on contextual similarities between words in an n-dimensional vector space. Nonetheless, it occasionally misrepresents the context. Consider the two sentences “I am going to the bank of Punjab to open an account” and “I went to the bank of the river at Punjab”. In these sentences, “bank” has two distinct meanings.

Some researchers [2] utilized the attention mechanism with RNN models, such as LSTM + attention and BiLSTM + attention, to overcome the constraints of RNN models, and they had some success. However, neither parallelism nor long-term dependencies are supported by the RNN. Therefore, authors in [3] employed the transformer model to detect hate speech on Twitter posts. However, their effort was restricted to English language texts.

Google has provided pre-trained BERT models and their variants based on transformer architecture for Natural Language Processing (NLP), including BERT, DistilBERT, RoBERTa, BERT-base-Chinese, XLM-RoBERT, and BERT Multilingual Cased. The pre-trained BERT models are trained on a vast quantity of data/corpus using powerful GPUs and have been implemented for various languages and domains. The BERT Multilingual Cased model has been trained on 104 languages; however, Roman Urdu is not included in 104 languages. 

According to our knowledge, there is no pre-trained BERT model for Roman Urdu text that can be fine-tuned for the classification task of Roman Urdu hate speech. Behind this motivation, we created a pre-trained BERT model (BERT-RU) trained on the biggest custom-built Roman Urdu dataset consisting of 173,714 text messages in the hate speech domain. In addition, we explored the transformer-based model for the Roman Urdu hate speech classification task, and its performance is compared to that of cutting-edge deep learning models. A cross-domain dataset is also used to evaluate the generalizability and robustness of the trained transformer model. The main contributions of this study are:1.To develop a first-ever Roman Urdu pre-trained BERT Model (BERT-RU), trained on the largest Roman Urdu dataset in the hate speech domain.2.To explore the efficacy of transfer learning (by freezing pre-trained layers and fine-tuning) for Roman Urdu hate speech classification using state-of-the-art deep learning models.3.To examine the transformer-based model for the classification task of Roman Urdu hate speech and compare its effectiveness with state-of-the-art machine learning, deep learning, and pre-trained transformer-based models.4.To show the robustness and generalization of the transformer-based model and other comparison models on a cross-domain dataset.

The remaining paper is organized as follows. Section 2 demonstrates related work to hate speech detection. Section 3 illustrates the methodology of the research. Section 4 describes experimental settings. Section 5 demonstrates results and discussions, and, finally, conclusions are drawn based on obtained results and then future directions are proposed.

## 2. Related Work

This section discusses prior research in the area of hate speech detection. Some studies employed a lexical method [4] to differentiate hate speech from other types of offensive language, but it was ineffective since it could only detect hate terms that the hate-based lexicon has classified. Traditional machine learning models were utilized by [5,6,7] for the automatic detection of hate speech. However, conventional machine learning models relied on human feature engineering, rendering them incapable of understanding text context as well as vulnerable to adversarial attacks [8]. Therefore, deep learning models were employed for the detection of hate speech. Deep learning algorithms automatically extract characteristics from input data. Some studies utilized a combination of deep learning models [9], while others used RNN models with attention, such as BiLSTM with attention [2]. Some researchers combined traditional and deep learning models such that LSTM, a deep learning model, was used for feature extraction, and GB Decision Tree, a traditional model, was utilized to accept those features as input and make predictions [10].

Based on languages, the literature on hate speech detection is divided into the following two sub-sections: 

### 2.1. English Hate Speech Detection

Most of the work in hate speech detection has been conducted in the English language, as English is the gold standard for Natural Language Processing (NLP). First, all novel techniques are applied to English language and then to other languages. A research by [11] proposed a lexicon-based approach for detecting English hate speech. The proposed method detects subjectivity in the sentences and builds a lexicon of hate-related words using a rule-based method. Finally, a classifier is trained on features extracted from the lexicon and tested on the documents to detect hate speech. The proposed method showed significant results. However, a significant shortcoming of this method is that it employs a lexical rule-based approach that disregards the domain and context of words in the text. The work by [4] observed that Lexical methods are effective in detecting potential offensive terms. However, it produced erroneous results in detecting hate speech, except for a few terms flagged by the hate-based lexicon that human coders identified.

Numerous studies showed that machine learning methods performed better than lexical methods for hate speech detection. The research by [4] proposed Logistic Regression with L regularization for hate speech detection in English tweets. They extracted unigram, bigram, and trigram features, and weighted them using the TF-IDF technique. The result showed that Logistic Regression performed substantially better than other models. Similarly, research by [8] proposed Logistic Regression with character-level features and showed that models trained on character-level features are more resistant to adversarial attacks than those trained on word-level features. However, the Logistic Regression may perform poorly on a huge dataset. Another research by [12] proposed a multi-view SVM technique with the added advantage of enhanced interpretability. They utilized multi-view SVM to capture a different aspect of hate speech within the classification process. The authors of [6] used a variety of feature extraction techniques and machine learning algorithms to determine which combination performed the best at automatic hate speech identification on public datasets. They observed that the Support Vector Machine (SVM), when used with bigram features weighted with TF-IDF, performed the best with an accuracy score of 79%, while KNN performed the worst. However, the suggested SVM model was incapable of real-time predictions and capturing the context of a text message. Nevertheless, the performance of classical models may diminish as the quantity of the dataset increases. The authors of [13] developed classifiers for hate speech and abusive language using multitask learning (MTL). They discovered that employing an MTL framework for detecting hate speech significantly improves a model’s capacity to generalize to new datasets. 

Detecting hate speech is a context-dependent challenge that requires a context-aware model. Nevertheless, based on feature engineering, traditional models could not efficiently understand the context of a text message. In addition, traditional models perform poorly with massive datasets. In contrast, deep learning models perform well on huge datasets and have the ability to comprehend the context of a text message. In this connection, the authors of [10] undertook several experiments with traditional ML methods, deep learning methods, and a combination of both to detect hate speech in Twitter posts. The empirical results demonstrated that the architecture of LSTM with Gradient Boosted Decision Tree (GBDT) earned the highest F1-score of 93% for the hate speech classification task. The architecture used LSTM for feature extraction and GBDT for hate speech classification in Twitter posts. However, the proposed architecture did not generalize well on cross-domain datasets since it used complete labelled data for feature extraction/embeddings before partitioning the dataset into training and test sets, which resulted in overfitting and an overestimated score. 

Research by [14] employed an unsupervised learning technique based on the Growing Hierarchical Self-Organizing map (GHSOM) to detect cyberbullying in social media. They employed hand-crafted features capable of capturing the semantic and syntactic properties of cyberbullying language. They chose two datasets from the FormSpring.me and YouTube platforms. The suggested method yielded average accuracy, precision, recall, and F1-score values of 0.69, 0.60, 0.94, and 0.74, respectively. However, the approach was incapable of identifying sarcastic messages. Similarly, the authors of [15] suggested a new unsupervised method for identifying hate speech on Facebook. They employed graph analysis to determine which websites were potentially disseminating hate speech. They used the K-Mean Clustering method to identify the most frequently discussed topics and compared them to hate speech. The outcome indicated that the proposed method attained an accuracy of 0.74. However, this work was limited to English hate speech and used English slang and slurs in the context of American society with diverse socioeconomic characteristics. However, the nature of hate speech varies based on language and other demographic characteristics. The slurs and derogatory terms in Roman Urdu differ from those in English.

On the other hand, research by [5] investigated different techniques for detecting abusive content. This study applied two deep learning techniques, namely the Convolutional Neural Network (CNN) and Recurrent Neural Network (RNN), to nine publicly available datasets. The SVM was also employed as a baseline classifier, with features extracted using the n-gram and average word2vec techniques. They found that word embedding trained on the same data source improved the prediction accuracy of deep learning models. The results demonstrated that deep learning models outperformed conventional SVM classifiers when the training dataset was severely unbalanced. However, when the class imbalance issue was addressed by oversampling, the performance of the SVM classifier improved dramatically and even surpassed that of deep learning models. Similarly, the authors of [16] conducted several experiments to detect cyberbullying on social media sites (SMPs). They used three huge datasets from Formspring, Twitter, and Wikipedia, as well as four Deep Neural Network (DNN) models, namely CNN, LSTM, Bidirectional LSTM (BiLSTM), and Bidirectional LSTM (BiLSTM) with Attention. The essential structure of all DNN models was identical to that of [10]. It was observed that the models preferred the non-bullying class since the datasets were fully unbalanced, with bullying comprising the minority class. In addition, it was found that oversampling significantly improved performance. In conclusion, they discovered that DNN models paired with transfer learning outperformed the state-of-the-art models on all three datasets. However, overfitting was a problem with this approach. The research by [17] proposed a dataset called ETHOS (online hate speech detection dataset) with two variants of data, i.e., binary label and multi-label. The proposed dataset is composed of text comments/reviews from YouTube and Reddit duly validated through crowd-sourcing. A variety of algorithms are employed to evaluate the proposed dataset in binary/multi-label scope in order to present the baseline performance on this dataset. It was observed that the performance of neural-based approaches was better than classical ML techniques. However, this research was limited to English comments, hence, the HS detection dataset was not versatile. 

During the COVID-19 era, the authors of [18] presented a text-based method for automatically detecting hate speech in online social networks. Empirical results demonstrated that the RNN outperformed both shallow and deep learning models in terms of accuracy on Dataset-1 and Dataset-2, respectively. In this study, however, the datasets were dispersed in an unbalanced manner. In addition, the study contributed to the detection of English hate speech posted on social media during the COVID-19 era but failed to detect Roman Urdu hate speech. 

The authors of [19] presented a transfer learning approach for detecting hate speech based on an existing pre-trained language model (BERT). They used two publicly available datasets to evaluate the suggested model. Next, they devised a mechanism for mitigating the impact of bias in the training set during the fine-tuning of a pre-trained BERT-based model for the hate speech detection task. However, this approach was limited to African American English (AAE) and Standard American English (SAE) languages and had not been tested on other cross-domain datasets containing diverse languages/dialects. Similarly, research by [20] utilized the BERT model for abusive language classification. They demonstrated that BERT outperformed other state-of-the-art models when fine-tuned for the underlying problem. However, fine-tuned BERT cannot be applied to other scripted languages, such as Roman Urdu, because there is no publicly available pre-trained BERT model for Roman Urdu.

### 2.2. Non-English Hate Speech Detection

Aside from the English language, some researchers have worked on hate speech detection in various other languages, such as [21,22], which employed a deep learning technique to detect hate speech in Arabic. The research by [21] created a public dataset of 9316 Arabic hate tweets and assessed four models such as CNN, gated recurrent units (GRU), CNN + GRU, and BERT. Empirical outcomes showed that the CNN model performed better than all other models in the same domain and cross-domain datasets. The results also demonstrated that the BERT model did not outperform the baseline and other models because BERT was pre-trained on Wikipedia and fine-tuned for Arabic tweets. Another study [22] used a cross-corpora multi-task learning model to detect Arabic hate speech. Two Arabic transformer-based models, AraBERT and MarBERT, were utilized in their proposed multi-task learning model. The experimental results demonstrated that the multi-task learning model was better than single-task learning in classification. The study of [23] suggested a pipeline to adapt the general-purpose RoBERTa language model to a text classification task, which was Vietnamese Hate Speech Detection (HSD). Initially, they tuned the PhoBERT on the HSD dataset by re-training the model on the Masked Language Model (MLM) task, then its encoder was used for text classification. The experimental findings showed that the suggested pipeline improved performance, establishing a new benchmark for Vietnamese Hate Speech Detection (HSD).

The authors of [24] suggested a multi-channel model (MC-BERT) with three BERT variants—English, Chinese, and multilingual BERTs—for detecting hate speech. In addition, they evaluated the use of translations as auxiliary input by translating training and test sets into the languages required by different BERT models. After being fine-tuned, the pre-trained BERT model attained state-of-the-art or comparable performance on three independent datasets. Roman Urdu, however, lacks a pre-trained BERT model and is extremely challenging to translate due to the lexical variances of Roman Urdu words. In addition, the translation process requires dictionaries in both languages; however, there is no Roman Urdu dictionary that provides the standard lexical form for Roman Urdu words. Similarly, the authors of [25] introduced TOCP, a huge dataset of Chinese profanity. The dataset consists of real sentences collected from social media sites, the profane expressions in those sentences, and suggestions for rephrasing those expressions less offensively while preserving their original meanings. Several neural network-based baseline systems were presented to evaluate the benchmark. The results indicated that neural network models performed better than other models. However, this study was limited to Chinese profanity detection. 

The authors of [26] employed the SVM-Radial Basis Function (RBF) classifier for detecting hate speech in Hindi–English mixed text on social media. They used a pre-trained word embedding technique, FastText, to extract text features. The effectiveness of the proposed method is compared with Word2vec and Doc2vec features, and it was discovered that FastText features provide a more accurate feature representation with the SVM-RBF classifier. The authors of [27] worked on hate and offensive speech detection in Hindi and Marathi. They utilized different deep learning architectures, including CNN, LSTM, and variants of transformer (BERT) models. The results indicated that transformer-based models performed the best, although CNN and LSTM with FastText embeddings yielded comparable results. In addition, they demonstrated that, with hyper-parameter tuning, the CNN and LSTM models outperformed the BERT models on the fine-grained Hindi dataset. 

The authors of [28] evaluated Hindi detection models for hate speech. To accomplish this evaluation, they employed a set of functionalities based on a real-world social media conversation. They considered Hindi as a base language and developed test cases for each functionality. They introduced the HateChekHIn evaluation dataset. To demonstrate the utility of these functionalities, they examined the cutting-edge transformer-based m-BERT model. 

Previously, few researchers have studied Roman Urdu hate speech detection on social media. The research of [7] assessed the efficacy of five supervised learning techniques, including deep learning, for detecting Roman Urdu hate speech. They also produced HS-RU-20, a corpus of 5000 Roman Urdu tweets. The results indicated that Logistic Regression with count vectorizer was the most effective algorithm for discriminating between hate and offensive tweets. The authors of [29] created the annotated dataset RUHSOLD, which contains 10,012 Roman Urdu tweets. They proposed a CNN-gram architecture with four CNN layers. In addition, they claimed the development of RomUrEm, a pre-trained BERT model for Roman Urdu. Still, it has not been made available to researchers so that they can assess its performance. However, CNN is unsuitable for sequential data such as text and ideal for image processing. The research of [30] created a manually annotated dataset for Urdu sentiment analysis and established baseline results by employing rule-based, machine learning, and deep-learning techniques. Moreover, they fine-tuned Multilingual BERT (mBERT) for Urdu sentiment analysis. The results demonstrated that the suggested mBERT model outperformed deep learning, machine learning, and rule-based classifiers and attained an F1 score of 81.49%.

The authors of [31] used transfer learning to detect Urdu hate speech on Twitter. The study contributed a labelled dataset, including 10,526 tweets in Urdu. They employed several ML algorithms as baseline models in conjunction with three text representation techniques, namely Count Vectorizer, TF-IDF, and Word2Vec. They discovered that Random Forest with count vectorizer outperformed other baseline models. They also employed transfer learning using pre-trained FastText Urdu word embeddings and Multilingual BERT embeddings to classify hate/offensive/neural speech. Lastly, they utilized the two variants of pre-trained BERT, xlm-ROBERTA and Distil-BERT. The findings indicated that these models were able to learn the context of tweets and effectively classify hate and offensive speech and, hence, achieved encouraging F1 scores. 

However, Roman Urdu is the name given to the Urdu language written using the Latin alphabet, often known as the Roman alphabet. It is written using the English alphabet based on the word’s pronunciation. Urdu itself is rarely utilized for hate speech detection. The Roman version of Urdu lacks resources. Therefore, this study attempted to develop a first-ever Roman Urdu pre-trained BERT Model (BERT-RU) and employed a transformer-based model for the classification of Roman Urdu hate speech.

## 3. Proposed Methodology

This section illustrates the essential architecture for detecting hate speech in Roman Urdu, as depicted in Figure 1. The architecture comprises the steps outlined below.

3.1. Data Selection.

3.2. Preprocessing.

3.3. Normalization.

3.4. Features Extraction/Embeddings.

3.5. Contextual Classification of Hate Speech using Transformer-based Model.

3.6. Training Phase.

3.7. Cross-Validation of Proposed Model.

3.8. Testing Phase.

Same-Domain Testing.Cross-Domain Testing.

### 3.1. Dataset Selection

In this research, we used the Roman Urdu Hate Speech Dataset (RU-HSD-30K) (https://github.com/Bilal4209/RU-HSD-30K.git, accessed on 23 October 2022) developed in our prior study [32]. The dataset (RU-HSD-30K) includes 30,000 text messages extracted from Twitter and Facebook, the two most prominent social media networks. The dataset includes two classes labelled “Hate” and “Neutral”. It is a balanced dataset since 15,000 messages are labelled “Hate” and 15,000 messages are labelled “Neutral”. Out of these 30,000 text messages, 26,000 were taken as a training dataset, and 4000 were chosen as a testing dataset. The training set was further divided into a training split and a validation split in a ratio of 80:20. During training, the validation split was used for cross-validation of the models. The generalization of all models was also evaluated on a cross-domain dataset developed for the sentiment analysis task [33].

### 3.2. Preprocessing

Data preprocessing is a vital step that employs a range of techniques to transform unprocessed data into a format acceptable for the ML model. In this study, the dataset was preprocessed in the usual manner, as is typical for nearly all NLP applications.

The (RU-HSD-30K) dataset contains text messages with one or several sentences per message. Initially, the dataset is converted into tokens by splitting the sentences into individual words using delimiters such as whitespace, tabs, semicolons (;), colons (:), commas (:), etc. For instance, “musalman saray dahshatgard hotay hain” is tokenized into “musalman”, “saray”, “dahshatgard”, “hotay”, and “hain”. White space is used to tokenize the preceding sentence.

Due to case sensitivity, “musalman” is treated differently than “Musalman” when extracting features from the data, which ultimately increases the number of unique words and makes the data complex and high-dimensional. Therefore, the data is transformed into a distinct format by being converted to lowercase.

Additionally, punctuation, symbols, and numerals are deleted during the preprocessing step because they are useless and cause the model to learn ineffective knowledge. Stop words are meaningless terms in the corpus, which do not contribute to the model’s effectiveness. These terms are eliminated from the dataset to minimize its dimensions. Typically, these terms contain prepositions, articles, conjunctions, etc.

### 3.3. Normalization

Roman Urdu is the name given to the Urdu language written using the English alphabet based on word’s pronunciation. Roman Urdu is not a standard language; therefore, it has no standard spellings for its vocabulary (words). Different individuals spell the same word differently in Roman Urdu. For instance, the word (زندگی) may be written as (zindagi, zindagee, zindge, zendagi, zendagee, etc.). Such lexical variations of words generate many unique terms that affect the model’s performance. These lexically variant words may be standardized to a single form.

There is no standard Roman Urdu dictionary (or spelling) to which word variants can be matched. In light of this, we used a 4000-word Roman Urdu dictionary compiled during our previous research [32]. The dictionary assisted in the standardization of lexical variations to a single form. Our prior research comprehensively explained the normalization of lexically variant terms/words [32]. In short, normalization of the Roman Urdu dataset is accomplished by matching the phonetic code of terms inside each group against the phonetic code of standard terms in the dictionary. If a match is found, then lexical variances within each group are replaced with the standard dictionary word, otherwise the words in the group remain unchanged [32]. The same procedure is applied to the entire dataset to replace variants of a term with its standard form.

### 3.4. Features Extraction/Embeddings

After preprocessing and normalization, the most crucial step is extracting features from raw data. The computer does not directly manipulate the raw data but transforms it into derived numerical values while maintaining the information in the original data. In conventional machine learning models, various feature extraction strategies, such as Bag of Words and TF/IDF (Term Frequency/Invert Document Frequency), along with n-grams and character grams, are frequently employed. Similarly, various embedding techniques, such as Word2Vec, GloVe, FastText, Elmo, and Pre-trained BERT Embeddings, are utilized in deep learning models.

In this study, the transformer-based model is employed, which has its own word embedding scheme called “TokenAndPositionEmbedding,” which consists of applying embedding to the input sequence (hate speech) and PositionEmbedding to the embedded tokens, and then summing these two results, thereby displacing the token embeddings in space to store their close, meaningful relationships.

### 3.5. Contextual Classification of Hate Speech Using Transformer-Based Model

Many traditional machine learning and deep learning techniques are available for Natural Language Processing (NLP) applications such as text classification. However, as described in Section 1 and Section 2, every technique has its own limitations. In this study, a transformer-based model is applied to the classification of hate speech to exploit its parallel processing capability and potential to capture the context of text messages.

This study applied the transformer-based model for the classification task of Roman Urdu hate speech. A team from Google Brain introduced the transformer model in 2017 [34]. Researchers are gradually replacing Recurrent Neural Network (RNN) models, such as Long Short Term Memory (LSTM), with the transformer model due to its more robust structure. The RNN model with an attention layer proved effective, but it still had problems with long-range dependencies, and its sequential nature prevented parallelization. In addition, the transformer allows simultaneous text processing, allowing it to be trained on an unparalleled quantity of information. This feature led to the creation of pre-trained models, such as Bi-Directional Encoder Representation from Transformers (BERT) and Generative Pre-trained Transformers (GPT). The transformer is developed to process sequence data such as natural language. It is used for Natural Language Processing (NLP) and Computer Vision.

This research developed a classifier model by leveraging a transformer block as a layer for the RU hate speech classification task. The architecture of the suggested classifier includes an input layer, an embedding layer, a transformer block, a Global Average Pooling, and a feed-forward network. The input text sequence is passed through the embedding layer, which performs token embedding and token position embedding. Token embedding turns every word of an input sequence into a 200-dimensional embedding vector. In positional embedding, each embedding vector representing an input word is augmented by summing it (element-wise) with a 200-dimensional positioning encoding vector, incorporating positional information into the input.

The augmented embedding vectors are fed into the transformer block layer consisting of self attention, normalization, and feed-forward networks. We used the TransformerBlock provided by Keras. The following configurations are made to train the proposed classifier based on the transformer. The maximum length of each input sequence is set to 200. The attention heads inside the transformer layer are set to 10. The hidden layer size for the feed-forward network inside the transformer layer is set to 32. The transformer layer produced one vector for each time step of our input sequence.

The transformer’s output is passed through a Global Average Pooling of one dimension, where the mean across all time steps is calculated. Finally, a feed-forward network is used to classify the input sequence/text as hate speech or neutral. The feed-forward network consists of a dense layer with a size of 20 with a ‘ReLU’ activation function, followed by a dropout layer with a 0.1 value. Finally, a final dense layer of size = 1 with a ‘Sigmoid’ activation function is used to classify the input sequence/text.

### 3.6. Training Phase

In our previous research, we created the Roman Urdu dataset RU-HSD-30K [32], which consists of 30,000 Roman Urdu text messages. The RU-HSD-30K dataset is split into training and test sets in the ratio of 87:13; hence, the model is trained on 26,000 RU text messages and tested on 4000 RU text messages. The training set of 26,000 RU text is further divided into training and validation sets in the proportion of 80:20. In other words, 20% of the training data is separated into a validation set. The suggested transformer-based model and other DL models are trained on the given dataset using binary cross-entropy as the loss function and Adam as the optimizer.

### 3.7. Cross-Validation of Proposed Model

As stated above, 20% of the training data is split into a validation set. A validation set is utilized during model training to adjust the model’s hyperparameters. Here, we employed the most basic form of cross-validation, known as held-out cross-validation. It effectively reduced bias because most of the data is utilized to fit the model. For the purpose of visualization, the “history” object is used to store the model’s accuracy and loss during training and cross-validation.

### 3.8. Testing Phase

In this phase, we performed two types of testing, the Same-Domain testing dataset and Cross-Domain testing dataset.

#### 3.8.1. Same Domain Testing

In this type of testing, the transformer-based model is evaluated using test data, composed of 4000 text messages, i.e., 13% of our dataset, RU-HSD-30K, and the outcome was recorded.

#### 3.8.2. Cross-Domain Testing

In this type of testing, the generalization of the transformer-based model is assessed using a Cross-Domain benchmark dataset from the UCI Machine Learning Repository [35].

## 4. Experimental Settings

In this section, the experimental settings for each model are specified. The dataset consisting of 30,000 text messages is classified as a same-domain dataset. At first, the dataset is preprocessed, tokenized, and normalized, then it is divided into a training set and testing set in the ratio of 87:13. For cross-validation, 20% of the training data is split into a validation set.

All the research experiments are conducted utilizing the Google-hosted Colab Pro Plus environment, which includes resources of Python 3, and Google Compute Engine Backend (GPU) with 85 GB of RAM, 200 GB of storage, and 500 compute units. All the deep learning techniques are implemented in Python using Keras-backed TensorFlow and TensorFlow. We utilized numerous Python libraries, such as Keras, PyTorch, Pandas, NumPy, NLTK, JSON, Gensim, and Sklearn.

### 4.1. Experimental Setting for Baseline (Traditional Machine Learning) Models

All the traditional ML models used the same experimental settings as in [32]. All the ML models were built, trained on the training dataset (26,000 text messages extracted from RU-HSD-30K), and evaluated on the testing dataset (4000 text messages taken out of RU-HSD-30K). The findings of ML models are directly taken from our earlier work [32] and served as baselines for this study.

### 4.2. Experimental Setting for Deep Learning Models

The experimental settings for DL models such as LSTM, BiLSTM, BiLSTM + Atten, and CNN were identical to those described in [32]. These models are trained on the training dataset (26,000 text messages taken from RU-HSD-30K) and evaluated on the testing dataset (4000 text messages from RU-HSD-30K). The outcomes of DL models are also obtained from our prior work [32] and served as baselines for this study.

### 4.3. Experimental Setting for Transformer-Based Model

A transformer is a deep learning model that utilizes the self-attention mechanism to weigh the importance of each component of the input data variably. The attention mechanism gives context for any position in the input data. The proposed transformer-based model is compiled with Adam, the optimizer, and Binary Cross Entropy, the loss function. The model is trained using a training dataset of 26,000 text messages taken from RU-HSD-30K. The trained model is then evaluated using a testing dataset containing 4000 text messages.

### 4.4. Experimental Setting for Transfer Learning

We employed two approaches for transfer learning. The first strategy employed embeddings of pre-trained models by freezing the pre-trained weights, while the second one involved fine-tuning/updating the weights of pre-trained models for the hate speech classification task.

#### 4.4.1. Transfer Learning by Using Pre-Trained Embeddings

In this strategy, the “trainable” parameter of the embedding layer is set to false so that embeddings from pre-trained models may be frozen during the training of the underlying models. We employed the pre-trained BERT models, specifically BERT-English, BERT-Multilingual, and BERT trained on RU from scratch. This work utilized the embeddings of these pre-trained models as-is with BiLSTM and BilSTM with Attention, to classify the RU Urdu hate speech.

#### 4.4.2. Transfer Learning by Fine-Tuning

In this method, the “trainable” parameter of the embedding layer is set to “True” so that pre-trained embeddings could be unfrozen and retrained during the training of the underlying models. We employed the pre-trained BERT models, specifically BERT-English, BERT-Multilingual, and BERT trained on RU from scratch. The embeddings of pre-trained models were fine-tuned in this work for RU hate speech classification. We fine-tuned the aforementioned pre-trained BERT models by coupling with different classifiers, including BiLSTM and BilSTM + Attention models, for classifying the RU hate speech.

### 4.5. Training Setup

The training section is divided into two parts:

4.5.1. Pre-Training of BERT on Roman Urdu Called BERT-RU.

4.5.2. Training of Underlying Models.

#### 4.5.1. Pre-Training of BERT on Roman Urdu (BERT-RU)

This section illustrates how BERT is pre-trained on a massive corpus of RU hate speech from scratch. Many pre-trained NLP models, such as Google’s BERT, are based on the transformer model. The BERT has been implemented in numerous domains and languages. Available BERT models include DistilBERT, RoBERTa, BERT-base-Chinese, XLM-RoBERT, and BERT Multilingual Cased, among others. The BERT multilingual base model (cased) is a BERT model that has been pre-trained on 104 languages, with a gigantic Wikipedia corpus using a masked language modelling (MLM) objective. Similarly, the BERT base model (cased) is another pre-trained model, trained on the English language.

However, there is no pre-trained BERT model on Roman Urdu available, which can be fine-tuned for the Roman Urdu hate speech classification task. Therefore, we performed BERT pre-training from scratch on a huge Roman Urdu dataset in this study. To achieve this purpose, we merged a training set consisting of 26,000 text messages from [32] with 147,714 text messages from the dataset of [36] by adjusting their class labels accordingly, resulting in a larger dataset of 173,714 text messages. The class labels for hateful and neutral speeches are “H” and “N,” respectively. The resulting corpus of 173,714 text messages is the first-ever largest Roman Urdu hate speech dataset. This largest dataset was further subdivided into a train set and a validation set, as shown in Figure 2 below.

The above train set, consisting of 156,342 samples, is used to train the Roman Urdu BERT, whereas the validate set consisting of 17,372 samples is used as a validation split. The following tokens are defined for the BERT tokenizer:

special_tokens = [

“[PAD]”, “[UNK]”, “[CLS]”, “[SEP]”, “[MASK]”, “<S>“, “<T>“]

Then, the WordPiece tokenizer is initialized as follows.

tokenizer = BertWordPieceTokenizer ( )

The tokenizer was configured and saved to a json file with the name “config.json”

After configuring the Tokenizer as shown in Figure 3, it is loaded as BertTokenizerFast. The sentences are passed through padding and truncation. Both training and testing datasets are tokenized. After that, input _ids and attention mask are set as PyTorch tensors. The batched is set to ‘True.’ The model is initialized with configuration. BERT Config () is provided with vocab_size and max_position_embedding. The configuration is provided to BertForMaskedLM( ). For masking the data collator, 20% of the tokens are randomly masked for Masked Language Modeling (MLM). The training arguments are configured as shown in Figure 4.

The trainer ( ) is initialized with values as shown in Figure 5.

The model is trained using the trainer.train() method. The trained model is saved to disc as checkpoints that may be utilized as a pre-trained BERT model named BERT-RU, a BERT model trained on the Roman Urdu hate speech dataset.

#### 4.5.2. Training of Underlying Models

For training and testing of transformer-based models, the dataset is divided into a training set and a testing set in the same manner, i.e., 26,000 messages are selected as the training set, and the remaining 4000 messages are picked as the testing set. The training set is subdivided into a training set and validation set in a ratio of 80:20, respectively.

The following experiments are conducted during the training of models for the RU hate speech classification task.

Experiment No.1: Transformer Model.Experiment No.2: BERT-RU + BILSTM (by Transfer Learning).Experiment No.3: BERT-RU + BILSTM (by Fine Tuning).Experiment No.4: BERT-RU + BILSTM + Attention (by Transfer Learning).Experiment No.5: BERT-RU + BILSTM + Attention (by Fine Tuning).Experiment No.6: BERT-English + BILSTM (by Transfer Learning).Experiment No.7: BERT-English + BILSTM (by Fine Tuning).Experiment No.8: BERT-English + BILSTM + Attention (by Transfer Learning).Experiment No.9: BERT-English + BILSTM + Attention (by Fine Tuning).Experiment No.10 BERT-Multilingual + BILSTM (by Transfer Learning).Experiment No.11 BERT-Multilingual + BILSTM (by Fine Tuning).Experiment No.12: BERT-Multilingual + BILSTM + Attention (by Transfer Learning).Experiment No.13: BERT-Multilingual + BILSTM + Attention (by Fine Tuning).

#### 4.5.3. Cross-Validation of Models

The training split, which consists of 26,000 text messages, is subdivided into a training split and a validation split in a ratio of 80:20, respectively. During the training of models, the validation split is utilized to adjust the model’s parameters. Here, we employed the most basic form of cross-validation, known as held-out cross-validation. The outcomes of each model during training and cross-validation are stored in the “history” object, which is then used for visualization.

### 4.6. Testing Setup

After training the models mentioned above on the training dataset (26,000 text messages), each model is then evaluated/tested on the same-domain testing dataset (4000 text messages), and the outcomes of each case are recorded. The performance of the models is assessed using well-known measures, namely accuracy, precision, recall, and F-score.

## 5. Results and Discussions

This section describes and concludes the outcomes of all experiments conducted in this study. The findings of classical machine learning and deep learning models are drawn from our past study [32], as shown in Table 1 of Section 4. In this research, the outcomes of these traditional ML and DL models served as baselines. In addition, the results of the proposed transformer-based model and pre-trained transformer-based models, as given in Table 2, are achieved after performing thirteen experiments. The experimental results presented in Table 1 and illustrated in Figure 6 indicate that XG Boost and Random Forest models performed better than other conventional models in terms of accuracy. In addition, the BiLSTM + Attention model outperformed both conventional and DL models [32].

We also employed transfer learning by combining the embeddings of pre-trained BERT models (BERT-English, BERT-Multilingual, and BERT-RU) with deep learning models for the task of RU hate speech classification. We also leveraged the possibility of retraining the weights of the existing pre-trained BERT models for the same classification task. As shown in Table 2, the empirical results demonstrate that the proposed transformer-based model performed better than pre-trained BERT models, including pre-trained BERT-English, pre-trained BERT-Multilingual, and even BERT-RU, for the classification of RU hate speech.

In addition, BERT-English + BiLSTM (Fine Tuning) outperformed all other transfer learning models. Figure 7 also visualizes the outcomes.

### 5.1. Discussion on Experimental Results

**Experiment#1:** The performance of the proposed transformer-based model was assessed using the four evaluation metrics of accuracy, precision, recall, and F-Score. We used visualization tools such as Python Matplotlib library to interpret the results.

The training trend depicted by the blue line in Figure 8 indicates that the accuracy of the proposed transformer-based model improves with each training epoch, beginning at 0.5759 and reaching 0.99. Likewise, the validation accuracy of the transformer-based model, depicted by the orange line, demonstrates a rising trend from 0.5090 to 0.9670.

Training accuracy and validation accuracy rise in a nearly identical manner. On the other hand, the training loss of the transformer-based model drops with each training epoch, beginning at 0.6875 and ending at 0.0275. Similarly, the validation loss of the same model decreases from 0.6936 to 0.1599. Hence, the training and validation losses are also lowered in roughly the same way. Therefore, it has been demonstrated that the proposed transformer-based model is a generalized model that may perform well on unseen data.

**Experiment#2:** In this experiment, we utilized transfer learning by freezing layers of pre-trained BERT-RU while training the model on the RU train set. The pre-trained BERT-RU embeddings are then given to BiLSTM to classify RU hate speech. The results are shown in Figure 9 and recorded in Table 2. The training trend depicted by the blue line in Figure 9 indicates that the accuracy of the BERT-RU + BiLSTM model increases with each training epoch, beginning at 0.5524 and reaching 0.9649. On the other side, the orange line represents the validation accuracy of the BERT-RU + BiLSTM model, which starts at 0.5623, reaches 0.8267, and then decreases to 0.8254. Overall, the BERT-RU + BiLSTM model does not reflect a good fit.

**Experiment#3:** In this experiment, the BERT-RU model is fine-tuned by training the entire model on the RU train set; thus, the model’s pre-trained weights are updated based on the RU dataset. The pre-trained BERT-RU embeddings are given to BiLSTM to classify RU hate speech. The results are recorded in Table 2 and shown in Figure 10.

The training trend depicted by the blue line in Figure 10 indicates that the accuracy of the BERT-RU + BiLSTM model grows with each training epoch, beginning at 0.670 and reaching 0.9204. Similarly, the validation accuracy of the BERT-RU + BiLSTM model, depicted by the orange line, demonstrates a rising trend from 0.7517 to 0.82.

On the other hand, the training loss of the BERT-RU + BiLSTM model reduces with each training epoch, beginning at 0.559 and decreasing to 0.190. However, the validation loss of the model falls from 0.5137 to 0.3984 before spiking. Overall, the model does not demonstrate a good fit.

**Experiment#4:** In this experiment, we leveraged transfer learning by freezing layers of pre-trained BERT-RU while training the model on the RU train set. The pre-trained BERT-RU embeddings are then given to the BiLSTM + Attention model to perform the RU hate speech classification task. The results are shown in Figure 11 and recorded in Table 2.

The training trend, depicted by the blue line in Figure 11, indicates that the accuracy of the BERT-RU + BiLSTM + Attention model increases with each training epoch, beginning at 0.5149 and reaching 0.9593. Likewise, the validation accuracy of the same model, depicted by the orange line, exhibits a rising trend from 0.5437 to 0.80. On the other hand, the training loss of the same model declines with each training epoch, beginning at 0.8041 and ending at 0.1123. However, the validation loss of the model starts at 0.6794, falls to 0.4825, and then steadily increases to 0.9425. Thus, the model does not show a good fit.

**Experiment#5:** In this experiment, fine-tuning of the BERT-RU model is accomplished by training the entire model on the RU train set; thus, the model’s pre-trained weights are updated based on the RU dataset. The pre-trained BERT-RU embeddings are then given to the BiLSTM + Attention model to classify RU hate speech. The results are shown in Figure 12 and recorded in Table 2.

The training trend depicted by the blue line in Figure 12 indicates that the accuracy of the BERT-RU + BiLSTM + Attention model grows with each training epoch, beginning at 0.6899 and reaching 0.9993.

In constrast, the validation accuracy of the BERT-RU + BiLSTM + Attention model, depicted by the orange line, begins at 0.8277, increases to 0.8671, and then gradually decreases to 0.8517. On the other hand, the training loss of the model declines with each training epoch, beginning at 0.6765 and ending at 0.0018. However, the validation loss of the same model decreases from 0.3955 to 0.3229 before gradually increasing to 2.33. Overall, the model appears to be overfitted.

**Experiment#6:** In this experiment, we exploited transfer learning by freezing layers of the pre-trained BERT-Multilingual while training the model on the RU train set. The pre-trained BERT-Multilingual embeddings are then given to BiLSTM to classify RU hate speech. The results are shown in Figure 13 and given in Table 2.

The training trend depicted by the blue line in Figure 13 indicates that the accuracy of the BERT-Multilingual + BiLSTM model increases with each training epoch, beginning at 0.6406 and reaching 0.9095. Likewise, the validation accuracy of the BERT-Multilingual + BiLSTM model, depicted by the orange line, exhibits a rising trend from 0.7079 to 0.8113. On the other hand, the training loss of the same model lowers with each training epoch, beginning at 0.672 and ending at 0.2176. However, the validation loss of the model declines from 0.5723 to 0.41824 before increasing to 0.4926. Overall, the model does not exhibit a good fit.

**Experiment#7**: In this experiment, the BERT-Multilingual model is fine-tuned by training the entire model on the RU train set; thus, the model’s pre-trained weights are updated based on the RU dataset. The pre-trained BERT-Multilingual embeddings are then fed to BiLSTM to classify RU hate speech. The results are shown in Figure 14 and recorded in Table 2. The training trend depicted by the blue line in Figure 14 indicates that the accuracy of the BERT-Multilingual + BiLSTM model grows with each training epoch, beginning at 0.8153 and reaching 0.9995.

In contrast, the validation accuracy of the BERT-Multilingual + BiLSTM model, depicted by the orange line, exhibits a trend beginning at 0.8608, increasing to 0.8721, and then gradually decreasing to 0.8465. On the other hand, the training loss of the same model lowers with each training epoch, beginning at 0.3988 and ending at 0.0023. However, the validation loss of the model exhibits a rising trend from 0.3201 to 1. Overall, the model is extremely overfitted.

**Experiment#8:** In this experiment, we explored transfer learning by freezing layers of the pre-trained BERT-Multilingual while training the model on the RU train set. The pre-trained BERT-RU embeddings are then given to the BiLSTM + Attention model to classify RU hate speech. The results are shown in Figure 15 and recorded in Table 2. The training trend depicted by the blue line in Figure 15 indicates that the accuracy of the BERT-Multilingual + BiLSTM + Attention model grows with each training epoch, beginning at 0.4999 and reaching 0.9087.

On the other hand, the validation accuracy of the same model, depicted by the orange line, exhibits a rising trend from 0.5223 to 0.80. However, the training loss of the BERT-Multilingual + BiLSTM + Attention model declines with each training epoch, beginning at 0.6955 and ending at 0.2396. The validation loss of the model drops from 0.6914 to 0.4753 before increasing to 0.5353. Overall, the model does not demonstrate a good fit.

**Experiment#9**: In this experiment, the fine-tuning of the BERT-Multilingual model is conducted by training the entire model on the RU train set; thus, the model’s pre-trained weights are updated based on the RU dataset. The pre-trained BERT-Multilingual embeddings are then fed to the BiLSTM + Attention model to classify RU hate speech. The results are shown in Figure 16 and recorded in Table 2. The training trend depicted by the blue line in Figure 16 indicates that the accuracy of the BERT-Multilingual + BiLSTM +Attention model grows with each training epoch, beginning at 0.6886 and reaching 0.9902.

On the other hand, the validation accuracy of the same model, depicted by the orange line, initially exhibits an increasing trend from 0.8273 to 0.8533 and then gradually drops to 0.8270. The training loss of the BERT-Multilingual + BiLSTM+ Attention model declines with each training epoch, beginning at 0.56 and ending at 0.029. However, the validation loss of the model initially falls from 0.4055 to 0.3644 before gradually increasing to 0.8362. Overall, the model displays exceptional overfit.

**Experiment#10:** In this experiment, we leveraged transfer learning by freezing layers of pre-trained BERT-English while training the model on the RU train set. The pre-trained BERT-RU embeddings are then given to BiLSTM for RU hate speech classification. The results are shown in Figure 17 and recorded in Table 2. The training trend depicted by the blue line in Figure 17 indicates that the accuracy of the BERT-English + BiLSTM model grows with each training epoch, beginning at 0.6119 and reaching 0.8715. Likewise, the validation accuracy of the model, illustrated by the orange line, exhibits a rising trend from 0.6854 to 0.82. However, the training loss of the same model falls with each training epoch from 0.6531 to 0.2992. Similarly, the validation loss of the model falls from 0.5965 to 0.4046. Thus, the model exhibits a good fit.

**Experiment#11:** In this experiment, the fine-tuning of the BERT-English model is accomplished by training the entire model on the RU train set; thus, the model’s pre-trained weights are updated based on the RU dataset. The pre-trained BERT-English embeddings are then fed to BiLSTM for RU hate speech classification. The results are shown in Figure 18 and recorded in Table 2.

The training trend depicted by the blue line in Figure 18 indicates that the accuracy of the BERT-English + BiLSTM model grows with each training epoch, beginning at 0.8168 and reaching 0.9991. In contrast, the validation accuracy of the same model, depicted by the orange line, starts at 0.8708, increases to 0.8725, and then gradually drops to 0.85. On the other hand, the training loss of the BERT-English+ BiLSTM model decreases with each training epoch, beginning at 0.3887 and ending at 0.0022. However, the validation loss of the model grows consistently from 0.2996 to 1.33. Overall, the model displays exceptional overfit.

**Experiment#12:** In this experiment, we leveraged transfer learning by freezing layers of the pre-trained BERT-English while training the model on the RU train set. The pre-trained BERT-RU embeddings are then given to the BiLSTM + Attention model for RU hate speech classification. The results are given in Figure 19 and recorded in Table 2.

The training trend depicted by the blue line in Figure 19 indicates that the accuracy of the BERT-English + BiLSTM + Attention model grows with each training epoch, beginning at 0.5210 and reaching 0.9878. Likewise, the validation accuracy of the BERT-English + BiLSTM + Attention model, depicted by the orange line, exhibits a rising trend from 0.5798 to 0.83. However, the training loss of the same model declines with each training epoch, beginning at 1.5774 and ending at 0.0394. The validation loss of the model diminishes from 0.7619 to 0.4082 before spiking to 1.2772. Hence, the model appears to be overfitted.

**Experiment#13:** In this experiment, the fine-tuning of the BERT-English model is accomplished by training the entire model on the RU train set; thus, the model’s pre-trained weights are updated based on the RU dataset. The pre-trained BERT-English embeddings are then fed to the BiLSTM + Attention model for RU hate speech classification. The results are visualized in Figure 20 and recorded in Table 2.

The training trend depicted by the blue line in Figure 20 indicates that the accuracy of the BERT-English + BiLSTM + Attention model grows with each training epoch, beginning at 0.6088 and reaching 0.9988. On the other hand, the validation accuracy of the same model, depicted by the orange line, starts at 0.8031, increases to 0.8785, and then gradually drops in each epoch until it reaches 0.8633. On the other hand, the training loss of the BERT-English+ BiLSTM + Attention Model drops with each training epoch, beginning at 1.5875 and ending at 0.005. In contrast, the validation loss of the model decreased from 0.4180 to 0.3222 before progressively increasing to 1.34. Overall, the model shows exceptional overfit.

The experimental results shown in Figure 7 and Figure 21 demonstrate that the proposed transformer-based model beats the classic ML and deep learning models, including LSTM, BiLSTM, CNN, and BiLSTM + Attention, in terms of accuracy and F-measure. Similarly, the suggested transformer-based model also outperformed pre-trained BERT models, such as BERT-English, BERT-Multilingual, and our own BERT-RU, which were coupled with DL models via transfer learning. The visualization results, as shown in Figure 8, reveal that the proposed transformer-based model exhibited superior performance in terms of accuracy and F-score. It also generalized well on cross-domain datasets compared to other pretrained models. The reason might be the transformer’s robust architecture, which is built on parallel processing and a self-attention mechanism that captures the context of text and gives more weight to relevant words. In addition, its parallel processing enables quick processing, allowing it to process the full input simultaneously.

In addition, the results demonstrated that BERT-RU performed the worst among transformer-based models for detecting Roman Urdu hate speech. In contrast, the pre-trained BERT-English model is a gold standard for English, but its performance remained consistent and superior to other pre-trained BERT models in the context of the RU hate speech classification task.

These findings suggest that the performance of the hate speech detection system is heavily dependent on the quality and quantity of the annotated training dataset. In addition, the results imply that models with fine-tuning are susceptible to overfitting in most cases. It might be because the weights of pre-trained models are updated during training on the hate speech dataset; as a result, the model memorizes the patterns in training data and cannot generalize effectively to new data.

### 5.2. Results of all Models on Cross Domain Dataset

The following evaluation metrics are used to assess the proposed transformer-based model and other deep learning models on cross-domain data to check their generalization, and their results are recorded in Table 3:Accuracy.Precision.Recall.F-Measure.

The above results indicate that the proposed transformer-based model outperformed all models on the cross-domain dataset. Likewise, BiLSTM + Attention also performed well on the cross-domain dataset.

## 6. Conclusions

Hate speech detection is a challenging problem that requires a robust application to detect and combat it in real-time. Hate speech detection is context-dependent and needs to be tackled with context-aware mechanisms. In this study, different ML and deep learning models, including LSTM, BiLSTM, BiLSTM + Attention, and CNN models, are used as baseline models in the context of the hate speech classification task. This study used the transformer-based model for RU hate speech classification due to its ability to capture the context of the hate speech text. We also used the power of BERT by pre-training it from scratch on the largest Roman Urdu dataset composed of 173,714 Roman Urdu messages.

In addition, we also leveraged transfer learning by coupling pre-trained BERT models with deep learning models. The evaluation metrics, such as accuracy, precision, recall, and F-measure, are employed to assess each model’s performance and the generalization of each model on a cross-domain dataset. Experimental results demonstrated that in terms of accuracy, precision, recall, and F-measure, the proposed transformer-based model outperformed traditional ML and other deep learning models, including LSTM, BiLSTM, and CNN, when directly applied to the classification of RU hate speech. Additionally, the suggested model exhibited improved generalization over a cross-domain dataset. The proposed model also showed superior performance as compared to pre-trained BERT models.

## 7. Future Directions

In this study, we constructed a pre-trained BERT for Roman Urdu text; however, its performance was inferior to that of pre-trained BERT-English and pre-trained BERT- Multilingual. We believe that there is a potential for improvement in the annotation guidelines established for Roman Urdu and the lexical normalization procedure for standardizing Roman Urdu terms.

Moreover, the results demonstrated that the BiLSTM + Attention model and the transformer-based model performed well in classifying RU hate speech. In future, we will evaluate the performance of the BiLSTM + Attention model by coupling it with other word embedding approaches, such as FastText, Glove, and Elmo. We also intend to apply the transformer-based model for detecting hate speech in Urdu, which is frequently used on social media by users from South Asia.

## Figures and Tables

**Figure 1 sensors-23-03909-f001:**
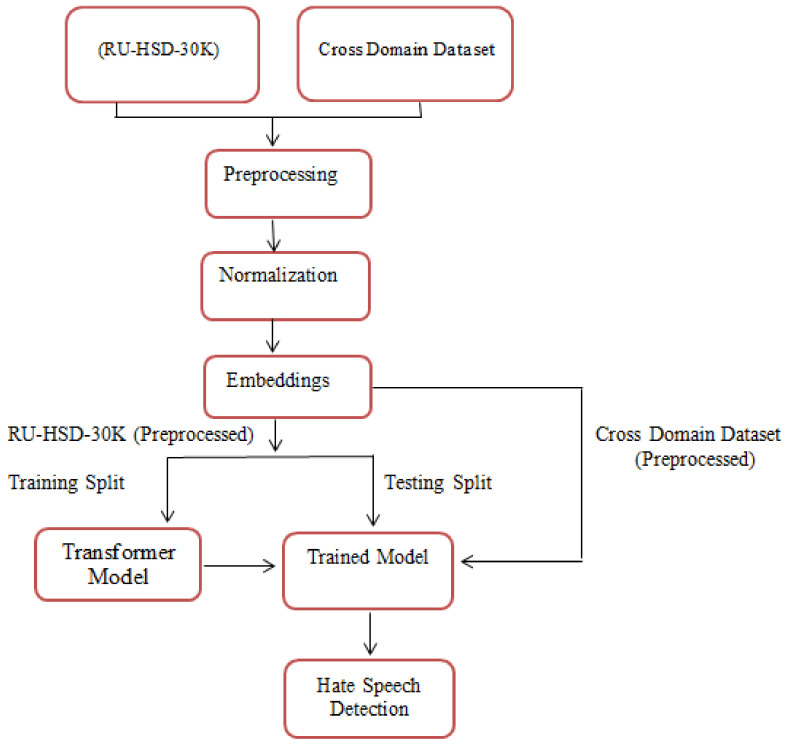
Proposed architecture for Roman Urdu hate speech detection.

**Figure 2 sensors-23-03909-f002:**
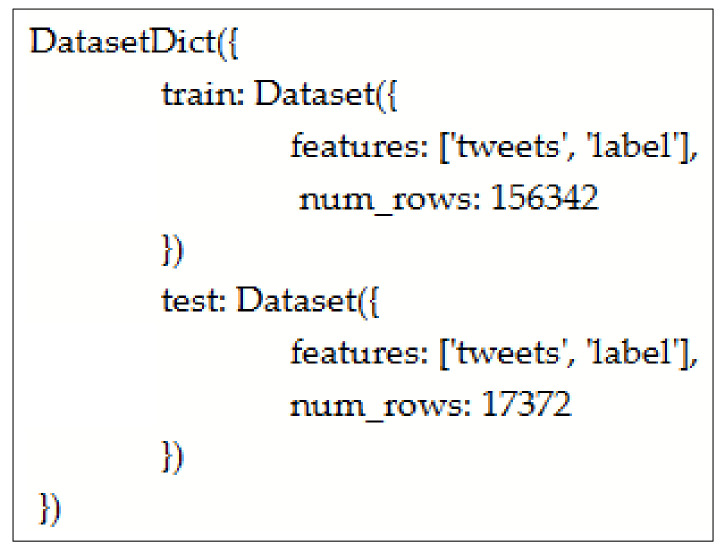
Division of dataset into train and validate splits.

**Figure 3 sensors-23-03909-f003:**
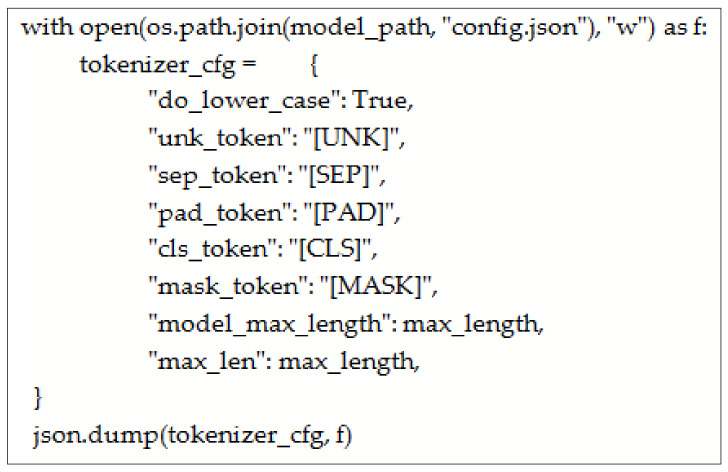
Configuration of BERT tokenizer.

**Figure 4 sensors-23-03909-f004:**
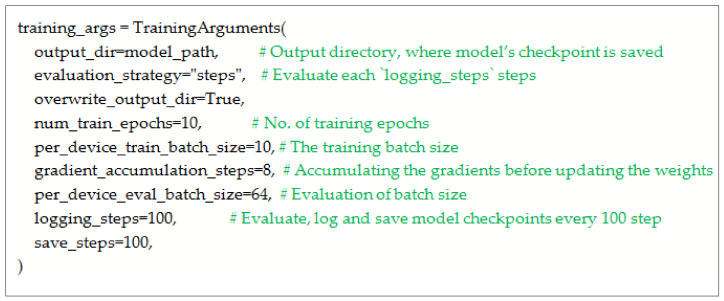
Configuration of training arguments.

**Figure 5 sensors-23-03909-f005:**
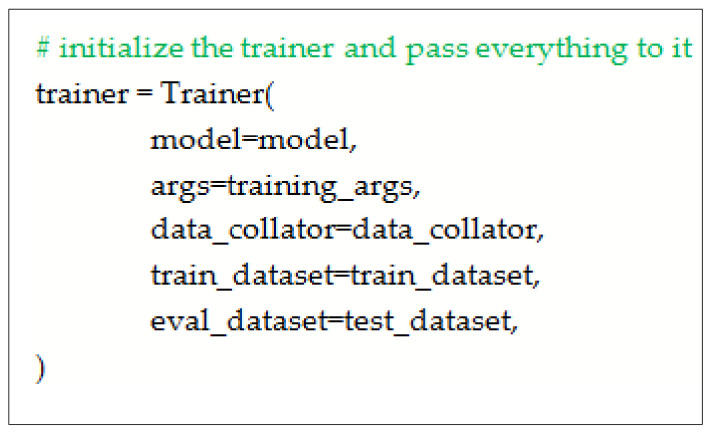
Configuration of training argument.

**Figure 6 sensors-23-03909-f006:**
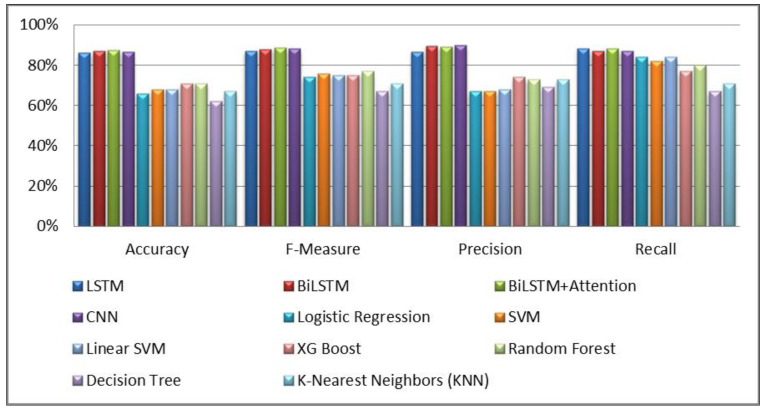
Comparison of deep learning vs. traditional models.

**Figure 7 sensors-23-03909-f007:**
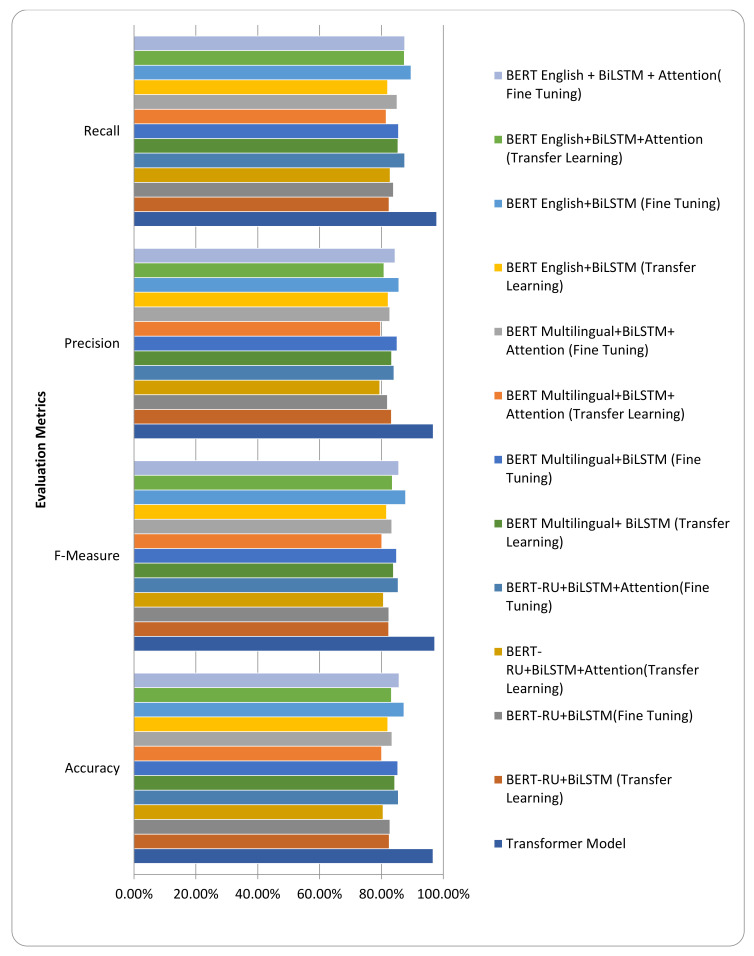
Comparison of transformer model vs. transfer learning models.

**Figure 8 sensors-23-03909-f008:**
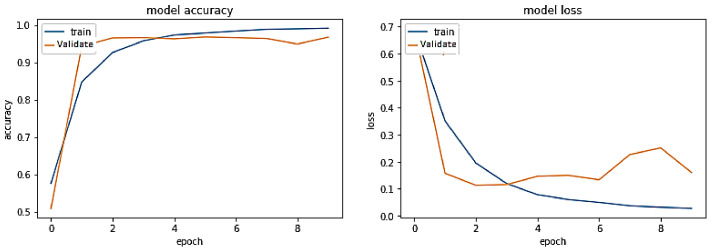
Performance of transformer model on the same-domain dataset.

**Figure 9 sensors-23-03909-f009:**
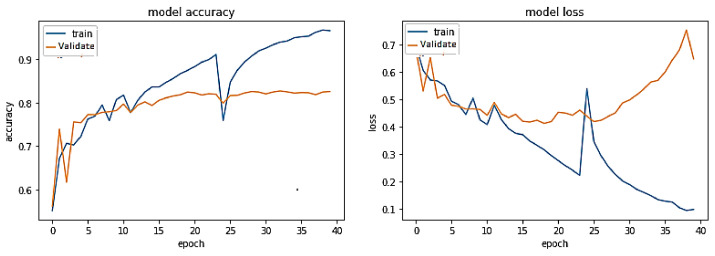
Pre-trained BERT-RU + BiLSTM (Transfer Learning).

**Figure 10 sensors-23-03909-f010:**
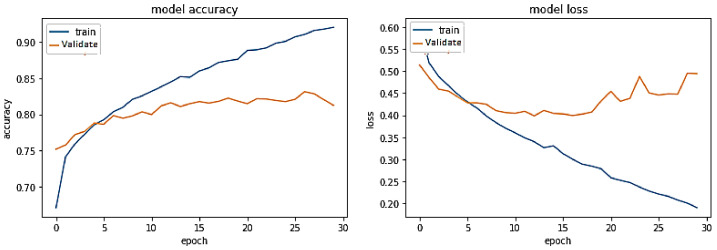
Pre-trained BERT-RU + BiLSTM (Fine Tuning) on the same-domain dataset.

**Figure 11 sensors-23-03909-f011:**
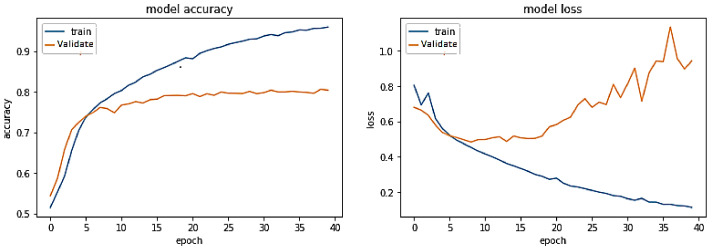
Pre-trained BERT-RU + BiLSTM with Attention (Transfer Learning) on the same-domain dataset.

**Figure 12 sensors-23-03909-f012:**
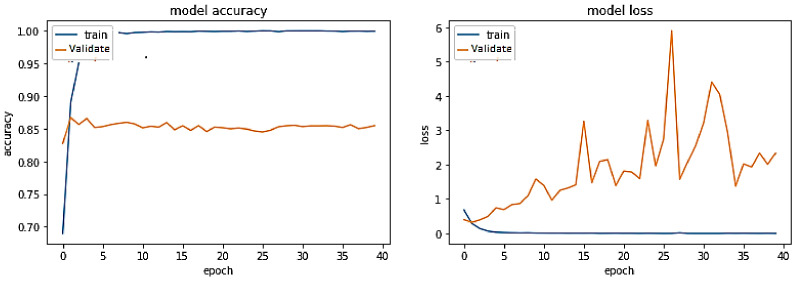
Pre-trained BERT-RU + BiLSTM with Attention (Fine Tuning) on a same-domain dataset.

**Figure 13 sensors-23-03909-f013:**
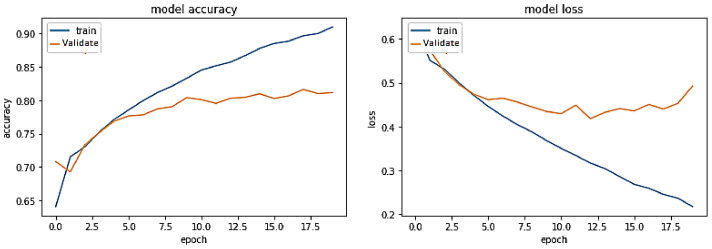
Pre-trained BERT-Multilingual + BiLSTM (Transfer Learning) on the same-domain dataset.

**Figure 14 sensors-23-03909-f014:**
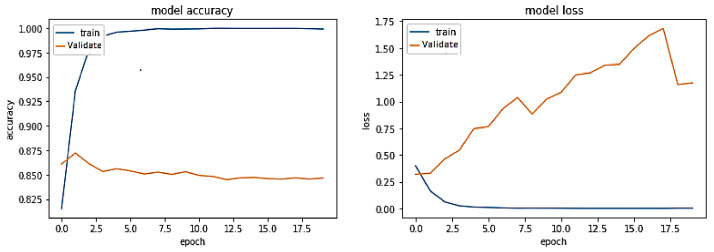
Pre-trained BERT-Multi + BiLSTM (Fine Tuning) on the same-domain dataset.

**Figure 15 sensors-23-03909-f015:**
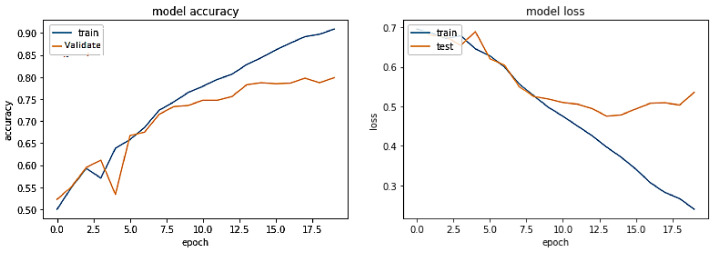
Pre-trained BERT-Multilingual + BiLSTM with Attention (Transfer Learning) on the same-domain dataset.

**Figure 16 sensors-23-03909-f016:**
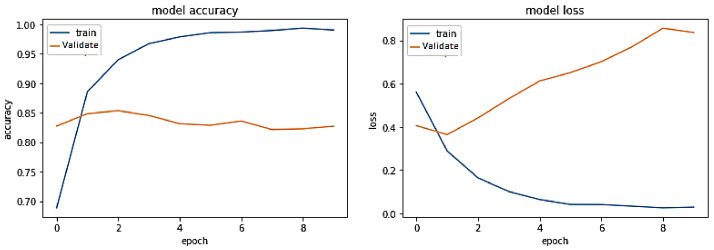
Pre-trained BERT-Multi + BiLSTM with Attention (Fine Tuning) on the same-domain dataset.

**Figure 17 sensors-23-03909-f017:**
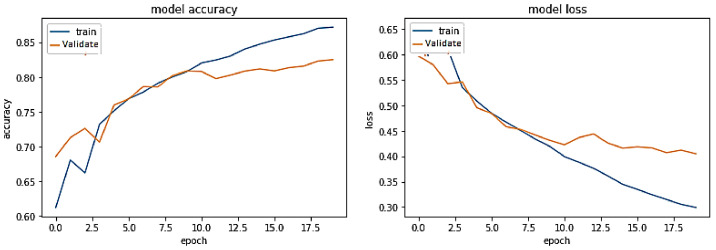
Pre-trained BERT-English + BiLSTM (Transfer Learning) on the same-domain dataset.

**Figure 18 sensors-23-03909-f018:**
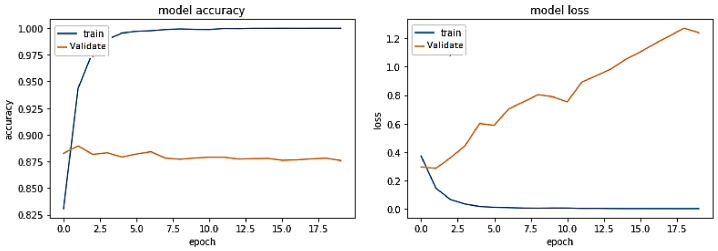
Pre-trained BERT-English + BiLSTM (Fine Tuning) on the same-domain dataset.

**Figure 19 sensors-23-03909-f019:**
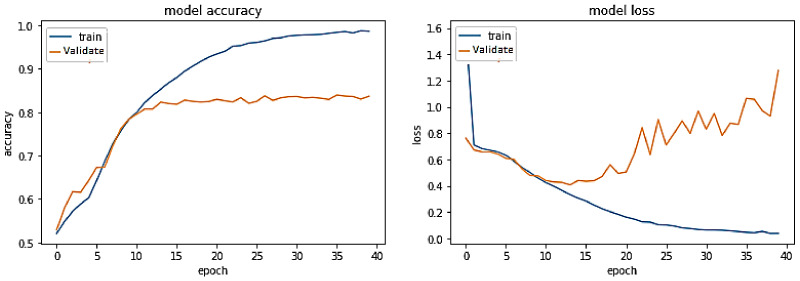
Pre-trained BERT-English + BiLSTM + Attention (Transfer Learning) on the same-domain dataset.

**Figure 20 sensors-23-03909-f020:**
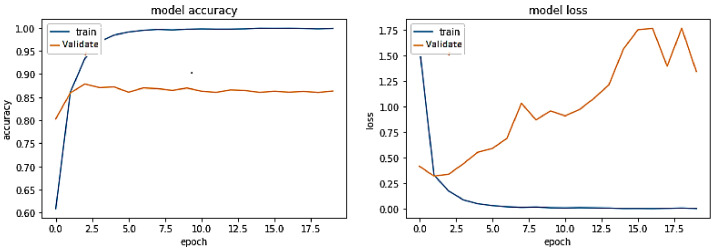
Pre-trained BERT-English + BiLSTM+Attention (Fine Tuning) on the same-domain dataset.

**Figure 21 sensors-23-03909-f021:**
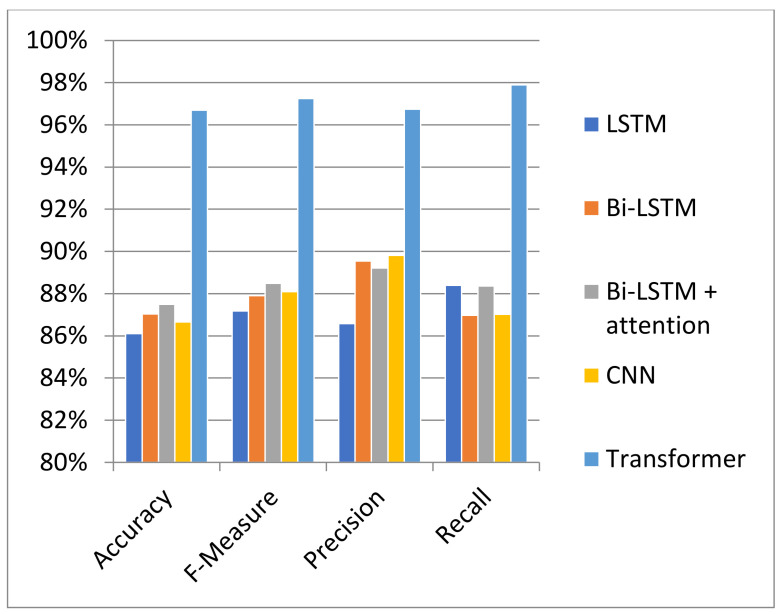
Comparison of deep learning models vs. transformer model.

**Table 1 sensors-23-03909-t001:** Results of traditional and deep learning models using word2vec embedding by [32].

Classifier Model	Accuracy	F-Measure	Precision	Recall
**Deep Learning Models**				
LSTM	86.11%	87.18%	86.58%	88.40%
BiLSTM	87.03%	87.91%	89.54%	86.97%
BiLSTM + attention	87.50%	88.48%	89.22%	88.36%
CNN	86.66%	88.10%	89.82%	87.02%
**Traditional Machine Learning Models**				
Logistic Regression	66%	74%	67%	84%
SVM	68%	76%	67%	82%
Linear SVM	68%	75%	68%	84%
XG Boost	71%	75%	74%	77%
Random Forest	71%	77%	73%	80%
Decision Tree	62%	67%	69%	67%
K-Nearest Neighbors (KNN)	67%	71%	73%	71%

**Table 2 sensors-23-03909-t002:** Results of transformer-based model and transfer learning from pre-trained BERT models.

Classifier Models	Accuracy	F-Measure	Precision	Recall
Transformer-based Model	96.70%	97.25%	96.74%	97.89%
Transfer Learning				
BERT-RU + BiLSTM (Transfer Learning)	82.53%	82.37%	83.22%	82.47%
BERT-RU + BiLSTM (Fine Tuning)	82.77%	82.43%	81.95%	83.88%
BERT-RU + (BILSTM + Attention) (Transfer Learning)	80.54%	80.66%	79.45%	82.81%
BERT-RU + (BILSTM + Attention) (Fine-Tuning)	85.46%	85.42%	84.08%	87.54%
BERT Multilingual + BiLSTM (Transfer Learning)	84.31%	83.86%	83.30%	85.35%
BERT-Multilingual + BiLSTM (Fine Tuning)	85.27%	84.86%	85.07%	85.51%
BERT-Multilingual + (BiLSTM + Attention) (Transfer Learning)	80.09%	80.13%	79.71%	81.55%
BERT-Multilingual + (BiLSTM + Attention) (Fine Tuning)	83.41%	83.36%	82.68%	85.08%
BERT-English + BiLSTM (Transfer Learning)	82.07%	81.63%	82.19%	81.99%
BERT-English + BiLSTM (Fine Tuning)	**87.29%**	**87.85%**	**85.66%**	**89.59%**
BERT-English + (BiLSTM + Attention) (Transfer Learning)	83.23%	83.55%	80.84%	87.39%
BERT-English + (BiLSTM + Attention) (Fine Tuning)	85.73%	85.58%	84.39%	87.54%

**Table 3 sensors-23-03909-t003:** Results of deep learning vs. transformer model (cross-domain dataset).

Classifier Models	Accuracy	F-Measure	Precision	Recall
LSTM	79.45%	78.49%	77.47%	81.05%
BiLSTM	80.16%	78.81%	77.15%	81.98%
BiLSTM + Attention Layer	80.90%	79.48%	78.02%	82.42%
CNN	79.90%	78.65%	76.96%	81.95%
Transformer Model	81.04%	79.64%	77.96%	82.82%

## Data Availability

Data is available on github link https://github.com/Bilal4209/RU-HSD-30K.git (accessed on 26 November 2022).

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
