# Peer review of "Roman Urdu Hate Speech Detection Using Transformer-Based Model for Cyber Security Applications"

_sensors, 2023, doi:10.3390/s23083909_

Round 1

Reviewer 1 Report (New Reviewer)

Title

Roman Urdu Hate Speech Detection Using Transformer Based Models for Cyber Security Applications

Review

This paper describes a methodology for determining whether text messages in Roman Urdu (that is, Urdu transliterated into the Latin alphabet) do or do not contain hate speech. I take as given that there is general consensus as to what qualifies as such, although granted there are gray areas.

In prior work the authors created a large dataset of text messages in Roman Urdu and applied several classification methods to determine which are hate speech and which are not. The dataset has 30000 messages, evenly split into "Hate" and "Neutral" groups. The test protocol uses 26000 for training/validation, with the remaining 4000 for testing. This paper carries forward that previous work by using new classifiers based on the BERT family of language embedders (roughly, neural networks trained to transform text into numerical vectors). Also in this work the authors have created a substantially larger dataset, using messages from an open source, and have applied their trained classifiers to this new corpus.

An important aspect of this and the previous work is to normalize language, using various tactics such as removal of stopwords, converting to lower case, converting common terms with variant spellings into one term and so forth.

The topic is important, the work is interesting and the methods seem appropriate. But this paper lacks clarity in several important respects and in particular it is possible that a validation set may have been misused as a test set. Until the various issues noted below are addressed it simply cannot be published.

The main issues involve explanation of what exactly was used for training, what for validation, what for testing, and what results apply to which sets. The paper uses three different datasets. Two were from the reference [33]) and from reference [35]. The third is from reference [37], and is used for cross-domain testing. It is not entirely clear what is in the training set. Section 3.6 notes different training protocols but less is stated about which were used in which specific experiments. It seems that the training is done both on the set from [33], using 26000 messages, and for the BERT methods it is also performed on a larger data set. Where is this larger set used? Is it only for the cross-validation test? If
not, were training and test texts kept separated? This is not at all clear. And it is important, because if they were not separated then the results are invalid.

Sections 4.5.1 and 4.5.2 do not clarify matters. The first mentions training on 156342 tweets and the second on 26000 messages. Is the first set exclusively from reference [35]? If so, it does not really make sense to compare results here to those in [33] because here the effective training set is substantially larger. Also there are 17372 tweets held out for testing. Was this intended to be as a validations set used internally by BERT? Or do these actually get used in later classification tests?

The focus of the paper is on what the authors refer to as the transformer model. But the experiments show 13 different classifiers, with most being BERT + various tuning variants. Were they all trained on the same set? In particular was the transformer model trained on the same set as the BERT-based classifiers? It is not really clear why the BERT classifiers are used in this work. Is it to provide a comparison methodology? This should be stated.

Section 5.2 is quite long and most of the BERT-based results could be summarized more succinctly. The bigger issue is that it is not clear what is being tested. The figures show "train" and "test" categories but the exposition mentions "train" vs. "validation" sets. But the test and validation sets are two different things and serve two different purposes. The validation set in particular is used in training to adjust tuning parameters.
The actual test set should only be used after training the classifiers is completed. Any other usage is not acceptable.

From the figures it seems that whatever set is being assessed is used during the training. I base this on the fact that the assessments are done following certain different training epochs and not only at the end of training. While this is useful for gauging accuracy and loss during training, it has no validity for purposes of actually testing the classifiers (they should never see the actual test set during training). So this point must be clarified by the authors. If in fact it was the validation set and not the test set that was used for these assessments, then the
results in table II are simply invalid. If instead the actual test sets were used, then it is quite unclear how information could have been obtained during the various training epochs.

Author Response

Reviewer 2 Report (New Reviewer)

The paper describes a solution for hate speach detection in urdu language, with  a novel bert model based machine learning solution.

major points:

- the quality of the text and the language is very mixed throughout the paper. Some paragraphs, like the literature review part, seem to provide a comprehensive literature review, where the first part of the chapter relies on the English sources and the other part of the paper checks the literature from other languages. But there are some huge editing mistakes the applied fonts and other formal criteria not followed. The pictures from line 612 - 613 suggests this is a very undemanding work. All of the figures and the applied algorithms shoud be referenced from the text, a caption should be added. This is necessary to do it.

- literature review, this is a strong part in the paper, this is deep but it  can be shortened a bit because all o the paper is very long, and some positive and negative feedback can be reformulated for the text referring to the main message of the given point more concretely. Some more examples from anonymization which can be a similar field, can be also added to represent some exotic languages (https://doi.org/10.3390/sym13081490, http://dx.doi.org/10.19044/esj.2019.v15n21p411).

- in the end of the introduction please reformulate the novelty description, it uses so much words which judging and sounds non-scientific it would be better to use some more concise sentences which shows the difference between the state of the art and the novel part of the paper, e.g the novel (first) bert model in urdu language

- it is necessary to rework the tables and the figures at the end of the paper and discuss them like figure 4, its hard to understand this image for first.

- instead of showing too much images please answer the question how is your results validated? how do you tested your code during the development?  this questions should be answered besides your impressive results.

Round 2

Reviewer 1 Report (New Reviewer)

This is much better. I found a handful of typographical/wording mistakes that should be corrected prior to publication.

(1) Experiment 7 line 767 claims "exceptionally fit". What is intended, I believe, is "exceptionally overfit" or "extremely overfit" (the term used for the same experiment in the original draft). Or could just say "overfit".

(2) Experiment 9 line 796 same issue as (1).

(3) Experiment 12 line 856 same issue again.

(4) Author Contributions line 943 has a wording issue. Perhaps "to the publish the" should be "to  publish the"

Author Response

Concerns of Reviewer#1: This is much better. I found a handful of typographical/wording mistakes that should be corrected prior to publication.

(1) Experiment 7 line 767 claims "exceptionally fit". What is intended, I believe, is "exceptionally overfit" or "extremely overfit" (the term used for the same experiment in the original draft). Or could just say "overfit".

(2) Experiment 9 line 796 same issue as (1).

(3) Experiment 12 line 856 same issue again.

(4) Author Contributions line 943 has a wording issue. Perhaps "to the publish the" should be "to  publish the"

Author response:

We are thankful to the reviewer for pointing out the typographical mistakes in the manuscript.

Author action:

We have addressed the reviewer’s concern in the updated manuscript.

Reviewer 2 Report (New Reviewer)

Dear Authors,

 Thank you for your corrections, but some of my major points are haven't answered:

-  the quality of some illustrations is poor: like from line 550-551, or line 560-561; these images without a caption and this low quality are unacceptable.

-  the quality of the used language is improved

- the text still looks like a cut and paste different formatting styles used along the paper

-  the quality of the images (I asked only to save higher quality and more nice colors in the same image) hasn't changed, like in Figure 2.

- validation: I asked how did you select the test/validation sets for hold-out training, the used architecture, and the used software; nothing personal, but to repeat your experiment, this information is necessary for interested readers.

Author Response

Reviewer#2, Concern # 1:

The quality of some illustrations is poor: like from line 550-551, or line 560-561; these images without a caption and this low quality are unacceptable.

Author response:

We thank the reviewer for drawing our attention towards the highlighted lines. The reported lines consisted of the research's code fragments, which were presented in image format. However, the reviewer's concern has been addressed, and all figures are now properly illustrated, captioned, and cited in the text.

Author action:  

We have addressed the reviewer’ comment in the updated manuscript. See Figures 2-4 for more detail.

Reviewer#2, Concern#2:  the text still looks like a cut and paste different formatting styles used along the paper.

Author response:

The reviewer’s concern has been addressed by using the formatting style of MDPI.

Author action:  

The reviewer’s concern has been addressed in the updated manuscript.

Reviewer#2, Concern # 3:

The quality of the images (I asked only to save higher quality and more nice colors in the same image) hasn't changed, like in Figure 2.

Author response:        

We worked diligently to improve the figures' quality. Now, all figures are legible and in a state of clarity.

Author action: We updated the manuscript by replacing the low-quality images with the improved ones.

This manuscript is a resubmission of an earlier submission. The following is a list of the peer review reports and author responses from that submission.

Round 1

Reviewer 1 Report

-   - The choice of tool (transformers) is well motivated. On the other hand, this has already proven the best choice for English, which is the “gold standard” for natural language processing (NLP). Comparison with the gold standard, at least by mentioning this in one sentence, should have been performed. The obtained results are more or less the expected ones considering the previous existing experience based on the gold standard of the field. Obviously, Roman Urdu is linguistically fundamentally different, so this computational similarity is worth mentioning and commenting.

-      - We appreciate cross-domain testing with sentiment analysis.

-  - Par. 3.2 should be compressed into only one sentence since what it describes is customary for all NLP applications. We suggest something of type: “Data was preprocessed in the usual way that is customary for all NLP applications: tokenization, conversion to lowercase, removal of punctuation, symbols and numbers, as well as stop words removal were performed”. (Contrary to that, it is good that normalization, which is typical of Roman Urdu, is discussed separately in par. 3.3).

-      - A reference (or more details concerning the work) should be given for the RU-HSD-30K corpus that is being used in the experiments and that apparently was developed by the authors prior to this work. The authors keep referring to their “previous work” without specifying what this work consists of, when it was performed etc. A reference would be needed here if it exists. Or other comments should be made.

-    - Minor but many English language issues, especially starting with par.5.2, but not only, should be fixed.

-   - More concerning the method itself: In the “Related Work” paragraph the authors stress the fact that most of the existing research concerning hate speech detection has been performed for English. This is natural because English is considered the gold standard for NLP. That is why novel techniques are usually first tested for English and then tested and/or adapted for another language. Very recent publications referring to hate speech detection in English have already declared transformer-based models as the intelligent baseline for this task. (See, for instance: Dascalu, S., Hristea, F., “Towards a Benchmarking System for Comparing Automatic Hate Speech Detection with an Intelligent Baseline Proposal”. Mathematics, 2022, 10(6), 945; https://doi.org/10.3390/math1006094). We are not surprised that transformer-based models worked best for Roman Urdu as well. Especially since the authors note on pp.3 the results of other authors showing that “pre-trained BERT developed for English text and retrained on Roman Urdu text using transfer learning beat BERT trained from scratch on Roman Urdu text”. Such remarks make it even more natural to think that what was indicated for English has chances of performing best for Roman Urdu as well and to compare the methodology and tools used for English as a gold standard to those used for Roman Urdu. Recent studies recommending transformer-based models as the intelligent baseline for this task use the BERT model in 3 architectural ways. The first and most basic way in which BERT can be employed is by using the BertForSequenceClassification function in PyTorch. The second possibility is using BERT’s output layer as an input to a multilayer perceptron or any other non-sequential network. Details concerning how training should be performed in this case are given in the previously mentioned paper, if not obvious. Finally, the third way in which BERT can be used is for embedding vector creation (see the same paper, at least). BERT can be used in combination with GloVe in two ways: by using it as an embedding vector or as a sequential model. It would be good for this paper to refine the discussion in this way, to replicate these experiments for Roman Urdu and to make specific recommendations for this language, according to the obtained results. Research of automatic hate speech detection is quite advanced in the case of the gold standard provided by English, so the results obtained so far for English should not be ignored when studying a different language, in this case Roman Urdu. The fact that transformer models, in general, are the best for this task, including for Roman Urdu, is an expected conclusion, considering the existing experience gained with hate speech detection so far. Refining the discussion in the suggested way and possibly coming to conclusions specific for Roman Urdu is, therefore, recommendable.

-    - Par. “7. Future Directions”: The paragraph should refer to future work for this specific topic, not for Roman Urdu NLP in general. For the task of hate speech detection we think that creating a general huge corpus of Roman Urdu, as suggested in the paper, is not what is needed. Rather it would be more useful to employ auxiliary datasets for this task, for example by creating a corpus for sentiment analysis in general or for offensive language in general. The mentioned recent study, for instance, comments that auxiliary datasets helped achieve higher accuracy if they have the same range of classification (sentiment analysis, offensive language etc.) and the same textual features (i.e. using Tweet datasets for longer text datasets can be detrimental to convergence). The creation of such a dataset for Roman Urdu would be more useful as future work concerning this topic.

I

Reviewer 2 Report

This manuscript presents a Transformer model for Roman Urdu Hate Speech Detection.

Strengths: The study objective is interesting.

Weaknesses:

1. Lack of novelty: There isn't much new technical information on the this task. The technical novelty is none or limited.
The Transformer method has been used in the previous works for the hate speech task.

2. Related work feels rather scarce. Please, insert more relevant references and include two subsections: English Hate Speech Detection and Non-English Hate Speech Detection.

3. Missing details with regards to dataset creation and method evaluation; the dataset origin is also not described in much detail, making it difficult for the reader to see the paper in an appropriate context.

4. In the Result section, the comparison with other works is limited.  Also, please consider more relevant baselines in the experiments, namely traditional machine learning techniques.

5. I believe that the Results section should also include a subsection for the Error Analysis and include samples accordingly.
Also, I believe that a section dedicated to hiper-parameter exploration and selection will be valuable. I would like to see why these specific parameters were chosen along with the authors' rationale for why their settings work best. What experiments led the authors to arrive at their specific combination.

6. Consider uploading your code to a public repository on Github and sharing the link.